

# Maturation trade-offs in octopus females and their progeny: energy, digestion and defence indicators

Alberto Olivares[1,*], Gabriela Rodríguez-Fuentes[2,3,*], Maite Mascaró[3,4], Ariadna Sanchez Arteaga[4], Karen Ortega[5], Claudia Caamal Monsreal[3,4], Nelly Tremblay[6] and Carlos Rosas[3,4]

[1] Departamento de Biotecnología, Facultad de Ciencias del Mar y Recursos Biológicos, Universidad de Antofagasta, Antofagasta, Chile
[2] Unidad de Química en Sisal, Facultad de Química, Universidad Nacional Autónoma de México, Sisal, Yucatán, Mexico
[3] Laboratorio Nacional de Resiliencia Costera (LANRESC), UNAM-CONACYT, Sisal, Yucatán, Mexico
[4] Unidad Multidisciplinaria de Docencia e Investigación, Facultad de Ciencias, Universidad Nacional Autónoma de México, Sisal, Yucatán, Mexico
[5] Posgrado en Ciencias del Mar y Limnología, Facultad de Ciencias, Universidad Nacional Autónoma de México, México, Ciudad de México, Mexico
[6] Biologische Anstalt Helgoland, Alfred-Wegener-Institut Helmholtz-Zentrum für Polar- und Meeresforschung, Helgolang, Germany
* These authors contributed equally to this work.

Corresponding author
Carlos Rosas, crv@ciencias.unam.mx

## ABSTRACT

Sexual maturation and reproduction influence the status of a number of physiological processes and consequently the ecology and behaviour of cephalopods. Using *Octopus mimus* as a study model, the present work was focused in the changes in biochemical compound and activity that take place during gonadal maturation of females and its consequences in embryo and hatchlings characteristics. To do that, a total of 31 adult females of *O. mimus* were sampled to follow metabolites (ovaries and digestive gland) and digestive enzyme activities (alkaline and acidic proteases) during physiological and functional maturation. Levels of protein (Prot), triacylglyceride (TG), cholesterol (Chol), glucose (Glu), and glycogen (Gly) were evaluated. Groups of eggs coming from mature females were also sampled along development and after hatching (paralarvae of 1 and 3 days old) to track metabolites (Prot, TG, Glu, Gly, TG, Chol), digestive enzymes activity (Lipase, alkaline proteases, and acidic proteases), and antioxidant/detoxification defence indicators with embryos development. Based on the data obtained, we hypothesized that immature females store Chol in their ovaries, probably from the food they ingested, but switch to TG reserves at the beginning of the maturation processes. At the same time, results suggest that these processes were energetically supported by Glu, obtained probably from Gly breakdown by gluconeogenic pathways. Also, was observed that embryos metabolites and enzyme activities (digestive and antioxidant/detoxification enzymes) where maintained without significant changes and in a low activity during the whole organogenesis, meaning that organogenesis is relatively not energetically costly. In contrast, after organogenesis, a mobilization of nutrients and activation of the metabolic and digestive enzymes was observed, together with increments in consumption of yolk and Gly, and reduction in lipid peroxidation.

Derived from our results, we also have the hypothesis that reactive oxygen species (ROS) were produced during the metabolic processes that occurs in ovarian maturation. Those ROS may be in part transferred to the egg provoking a ROS charge to the embryos. The elimination of ROS in embryos started when the activity of the heart and the absorption of the yolk around stages XIV and XV were evident. Altogether, these processes allowed the paralarvae to hatch with buffered levels of ROS and with the antioxidant defence mechanisms ready to support further ROS production derived from paralarvae higher life stage requirements (feeding and metabolic demands).

## INTRODUCTION

Sexual maturation and reproduction influence the status of a number of physiological processes and consequently the animal's ecology and behaviour (*Zamora & Olivares, 2004*). Earlier studies realized in four cephalopods species indicated energy acquisition for egg production directly from food rather than from stored products in digestive glands (DGs) or muscle (*Farías et al., 2011*; *Rosa, Costa & Nunes, 2004*). This finding highlighted the importance of trophic ecology for optimal reproduction of those species, awaking the need for laboratory studies assessing the influence of each type of food on the biochemical characteristics of octopus eggs, embryos and hatchlings (*Márquez et al., 2013*; *Steer et al., 2004*). Studies made in *Octopus maya* demonstrated that the type of diet (fresh or formulated) during female maturation has influence on biochemical and morphological characteristics of embryos and hatchlings (*Caamal-Monsreal et al., 2015*; *Tercero-Iglesias et al., 2015*): octopus females fed with mixed diets (squid and crab; crab and fish) provide a higher yolk quality, allowing hatchlings a better performance during the first days of culture, in comparison with hatchlings from females fed with a mono diet (only crab, squid, or fish). So, the health status of *O. maya* females derived of their nutritional condition strongly influences the capacity of the animals to maintain their physiological integrity during post-spawning period, when female need to protect the eggs during the embryo development (*Roumbedakis et al., 2017*). In *O. vulgaris*, mixed maternal diet (crab and fish) provoked more hatchlings than mono diet (only crabs) (*Márquez et al., 2013*). This suggests that the relationship between the nutrients of the diet (amino and fatty acids) and the embryo characteristics previously observed in *O. maya* (*Caamal-Monsreal et al., 2015*) could be encountered in other octopus' species. The way nutrients are processed in females during maturation, stored in eggs and used afterward by embryos for their development still remain unknown.

Recent studies have demonstrated steep physiological changes related with octopus embryo development, influencing the form in which nutrients stored in yolk are used to synthesize organs and tissues when embryos are exposed at different temperatures.

*Caamal-Monsreal et al. (2016)* and *Sánchez-García et al. (2017)* showed that from stage XV onwards (after the completion of organogenesis), the yolk consumption of *O. maya* embryos was significantly higher than observed in previous stages indicating an acceleration of embryo metabolism to stimulate growth. From the same stage, there is an increment in catabolic enzymes activity that transform yolk in molecules physiologically useful for embryos (*Caamal-Monsreal et al., 2016*). That reserve mobilization is probably provoking, together with increment in oxygen consumption, higher production of reactive oxygen species (ROS), which at the end are temperature dependent (*Repolho et al., 2014*; *Sánchez-García et al., 2017*). In ectothermic organisms, as a result of aerobic metabolism endogenous ROS production (e.g. superoxide, peroxyl radical, and hydroxyl radicals) could lead to oxidative stress (*Regoli et al., 2011*; *Vinagre et al., 2014*). Oxidative stress can damage lipids, proteins (Prot), and DNA, and cause impaired cellular function or apoptosis (*Regoli & Giuliani, 2014*). To prevent oxidative stress and keep the balance of the cellular redox state, aerobic organisms have evolved an efficient antioxidant defence system that consists of both non-enzymatic small antioxidant molecules (e.g. reduced glutathione (GSH), ascorbic acid, carotenoids, etc.) and a cascade of antioxidant enzymes (*Regoli & Giuliani, 2014*). Until now, no efficient ROS elimination antioxidant mechanisms have been identified in octopus embryos exposed to thermal changes (*Repolho et al., 2014*; *Sánchez-García et al., 2017*). For that reason, a solid base line providing the links between the use of yolk reserves and the detoxification/antioxidant defence mechanisms during octopus embryos development is essential to understand the possible defence responses against future environmental stressors.

In *O. tehuelchus*, *Fassiano, Ortiz & Ríos De Molina (2017)* find that lipid peroxidation (LPO) in the female ovaries is reduced with maturation, despite no changes in the antioxidant enzyme superoxide dismutase (SOD) activity or in GSH levels. In oviducal gland, SOD activity and LPO levels increased during maturation, while the highest values of GSH were recorded after the spawn. Based on these results and taking into account that lipid reserves are involved in yolk production during vitellogenesis, we could hypothesize that part of the oxidized lipids in female could be placed into the yolk during the egg formation eventually affecting the ability of embryos to eliminate ROS, as was observed when oxidative stress is provoked by high temperatures in the growing phase (*Repolho et al., 2014*; *Sánchez-García et al., 2017*).

Our model, *O. mimus*, is one of the most important octopus species of the Pacific Ocean at the South of Ecuador, with a distribution from Callao (Peru) to San Vicente bay (Chile), sustaining important artisanal benthic fishing grounds in both countries (*Cardoso, Villegas & Estrella, 2004*; *Cortez, Gonzalez & Guerra, 1999*; *Olivares et al., 1996*). The species reproduces throughout the year, with one or two seasonal peaks (marked by mature females) that are specific for each country (*Cardoso, Villegas & Estrella, 2004*; *Castro-Fuentes et al., 2002*; *Olivares et al., 1996*). One individual egg laying can extend over 20 days due to asynchrony in oocyte development and the loss of ovarian function attributable to their semelparous reproductive strategy (*Zamora & Olivares, 2004*). The time required for embryonic development changes with environmental temperature: 67–68 days in winter (16 °C) and 38–43 days in summer (20 °C)

(*Castro-Fuentes et al., 2002*; *Warnke, 1999*). Many parameters have been investigated to enhance the spawns in laboratory conditions: 1) type of tanks, light and feeding (*Cortez, Castro & Guerra, 1995*; *Olivares et al., 1996*; *Zuñiga, Olivares & Ossandón, 1995*); 2) histology, biochemistry and reproduction (*Cortez, Castro & Guerra, 1995*; *Olivares et al., 2017*; *Olivares et al., 2003*; *Olivares et al., 2001*; *Zamora & Olivares, 2004*); 3) embryo development (*Castro-Fuentes et al., 2002*; *Uriarte et al., 2012*; *Warnke, 1999*); 4) thermal tolerance of paralarvae (*Zuñiga et al., 2013*); 5) growth and nutrition (*Baltazar et al., 2000*; *Carrasco & Guisado, 2010*; *Gallardo et al., 2017*); and 6) taxonomy and genetics (*Perez-Losada, Guerra & Sanjuan, 2002*; *Söller et al., 2000*). All those studies were stimulated in the view of aquaculture (minimum requirements for maximum yield) and to assess the potential effects of environmental changes in the natural populations (*Zuñiga, Olivares & Rosas, 2014*).

In an attempt to define the relationship between physiological characteristics of cephalopod females, and the physiology of embryos and hatchlings of *O. mimus,* the present study was designed to understand the changes that take place during gonadal maturation of *O. mimus* females (energetic metabolites and digestive enzymes) and their effects in embryo and paralarvae characteristics (energetic metabolites, digestive enzymes, antioxidant/detoxification defences). A base line of those processes was constructed and hypothesis were proposed to explain if maturation process into females affects the physiological condition of octopus embryos during its development. To do that we characterised: 1) the maturation processes of females both morphologically (maturity stages) and biochemically (energetic metabolites in ovaries and DG together with digestive enzymes in the latter); 2) the embryo biochemical changes along development including energetic metabolites, digestive proteinases and lipases (as yolk consumption proxy), and some enzymes involved in the detoxification/antioxidant defence mechanisms. Also, LPO and total GSH were analyzed as proxies of oxidative stress. Those evaluations were also done in hatched paralarvae (PL) after one (PL1) and three (PL3) days.

## MATERIAL AND METHODS

The actual study was approved by the Animal Care committee of the Universidad de Antofagasta, Chile and followed the Experimental Animal Ethics Committee of the Faculty of Chemistry at Universidad Nacional Autónoma de México (Permit Number: Oficio/FQ/CICUAL/100/15).

### Animals

A total of 31 females *O. mimus* (body weight-BW of 1,179 ± 651 g; mean ± standard deviation) were collected by scuba diving and using the gear hook (the local method), between one and five m depth, off Antofagasta, Chile (23°38′39 S, 70°24′39 W). All captured females above these sizes have a developed reproductive system (*Zuñiga, Olivares & Rosas, 2014*).

### Reproductive condition

In the laboratory, 23 females were immersed in cold sea water (6–8 °C), as recommended for sub-tropical cephalopod species (*Sweeney & Roper, 1983*), to induce loss of sensation

and enable humane killing (*Andrews et al. 2013*). Soon after dormancy, each octopus was weighed (BW ±0.001 g) and dissected to obtain the mass (±0.001 g) of reproductive and digestive organs: ovaries weight (OW); reproductive system weight (RSW; composed of the ovaries with oviducts and oviducal glands); and DG weight. DG and ovaries samples were placed in Eppendorf tubes and freeze at −80 °C. After dissection, eviscerated body weight (EBW) was also obtained. From this information, several indices were calculated:

Reproductive system weight index $(RSWI, \%) = (RSW/BW - RSW) \times 100$

Gonadosomatic index $(GI, \%) = (OVW/BW - OVW) \times 100$

Digestive gland index $(DGI, \%) = (DG/BW - DG) \times 100$

A histological approach was used posteriorly to classify the maturation status of the females (*Avila-Poveda et al., 2016*; *Olivares et al., 2017*). Preserved ovary samples (one per each sampled female, $n = 23$) were cut through the middle level into transverse and longitudinal sections and washed in 70% ethanol. The sections were then dehydrated in ethanol series, cleared in benzene, and infiltrated and embedded in paraplast®. Serial sections were cut at a thickness of five μm using a manual rotary microtome (Leitz 1512, Wetzlar, Germany), and were mounted on glass slides and stained using the routine Harris haematoxylin-eosin regressive method ($HHE_2$; *Luna, 1968*, *Howard et al., 2004*). Alcian blue at pH 1.0 was used to contrast acidic mucopolysaccharides (*Humason, 1962*).

## Embryo development

The remaining females ($n = 8$) were individually placed in 108 L tanks with open and aerated seawater flow in a semi-dark environment to stimulate the spawn, at optimum temperature of 16–20 °C (*Uriarte et al., 2012*). Spawn occurred after 8–20 days. A rope of eggs from each female was sampled every 4–7 days. Each embryo was classified by stage of development according to *Naef (1928)* and individually stored at −80 °C for biochemical analysis. Stages of *O. mimus* paralarvae of 1 and 3 days old were also individually preserved. Those paralarvae were not feed. At the end of the experiment, all preserved samples were freeze dried to facilitate the transportation of samples to the Unidad Multidisciplinaria de Docencia e Investigación (Faculty of Science, Universidad Nacional Autónoma de México, Sisal, Mexico) for their biochemical analysis.

## Biochemical analysis in females, embryos and paralarvae

### Metabolites (females, embryos and paralarvae)

Samples were diluted 1:10 (w/v) and homogenized in cold buffer Tris pH 7.4 using a Potter-Elvehjem homogenizer. Homogenized samples were centrifuged at 10,000 *g* velocity for 5 min at 4 °C, and the supernatant was separated and stored at −80 °C until analysis. Metabolites were analyzed with commercial kits following manufacturer's instructions (ELITech, Paris, France): triacylglyceride (TG; TGML5415), cholesterol (Chol; CHSL5505) and glucose (Glu; GPSL0507). Supernatants were diluted 1:300 for soluble Prot

determination using a commercial kit (Cat. 500-0006; Bio-Rad, Hercules, CA, USA) (*Bradford, 1976*). All the absorbances were recorded using a microplate reader (Benchmark Plus, Bio-Rad, Hercules, CA, USA) and concentrations were calculated from a standard substrate solution and expressed in mg.mL$^{-1}$. Glycogen (Gly) was determined using the method described by *Carroll, Longley & Roe (1956)* and expressed in mg.g$^{-1}$.

### Digestive enzyme activity assays (females, embryos and paralarvae)

Alkaline proteases activity of the extracts was measured at pH 8 with the method of *Kunitz (1947)* modified by *Walter (1984)*, using 1% (w/v) casein (1082-C Affymetrix, Santa Clara, CA, USA) as substrate in 100 mM universal buffer. Acid proteases activity was evaluated at pH 4 according to *Anson (1938)* with adjustments, using a solution of 1% (w/v) haemoglobin (hammarsten quality 217500; BD-Bioxon, Mexico) in universal buffer (*Stauffer, 1989*). In both assays, 0.5 mL of the substrate solution was mixed in a reaction tube with 0.5 mL of universal buffer and 20 μL of enzyme preparation (1:100 sample dilution) and incubated for 10 min at 35 and 40 °C for alkaline proteases and acid proteases, respectively. Then, the reaction was stopped by adding 0.5 ml of 20% (w/v) trichloroacetic acid and by cooling on ice for 15 min. The undigested substrate precipitate was separated by centrifugation at 13,170 $g$ for 15 min. The absorbance of the supernatants was measured spectrophotometrically at 280 nm. Control assays (blanks) had trichloroacetic acid solution before the substrate was added. Lipase activity was measured in microplate according to *Gjellesvik, Lombardo & Walther (1992)* using 200 μL substrate solution (Tris(hydroxymethyl)aminomethane or Tris 0.5 M, pH 7.4, sodium taurocholate five mM, sodium chloride 100 mM, four nitrophenyl octanoate 0.35 mM) and five μL enzyme preparation (1:2 sample dilution in TRIS 0.5 M, pH 7.4). Absorbance was read at 415 nm, every minute, during 10 min. All digestive enzyme activities were expressed as activity unit (U; change in absorbance per minute) per milligram of Prot. Lipase activity was only measured in embryos and paralarvae of one and three days old.

### Detoxification and oxidative stress indicators (embryos and paralarvae)

Lyophilized samples were homogenized in cold buffer Tris 0.05 M pH 7.4 at 1:100 (w/v) using a Potter-Elvehjem homogenizer (Thomas Scientific, Swedesboro, NJ, USA). Redox potential ($E_h$) was measured with a probe (ORP-146CXS; ArrowDox Measurement System, Los Angeles, CA, USA) in each homogenate (in mV). Posteriorly, homogenate was divided for triplicate assays of activities of acetylcholinesterase (AChE), carboxylesterase (CbE), catalase (CAT), glutathione S-transferase (GST), and for levels of LPO and total GSH. The homogenates were centrifuged at 10,000 $g$ for 5 min at 4 °C for the enzymatic assays. The supernatants were separated and stored at −80 °C until analysis, together with the homogenates for LPO and GSH. All the enzyme activities were analyzed at 25 °C and reported in activity unit (U; nmol per minute) per milligram of Prot.

AChE is an enzyme that catalyzes the breakdown of acetylcholine and other choline esters. This enzyme is mainly found at neuromuscular junctions and in synapses. AChE activity was measured using a modification of the method of *Ellman et al. (1961)* adapted to a microplate reader (*Rodríguez-Fuentes, Armstrong & Schlenk, 2008*). Each well

contained 10 μL of the supernatant and 180 μL of 5, 5′ -dithiobis (2 nitrobenzoic acid) 0.5 M in 0.05 mM Tris buffer pH 7.4. The reaction started by adding 10 μL of acetylthiocholine iodide (final concentration one mM) and the rate of change in the absorbance at 405 nm was measured for 2 min. CbE are ubiquitous enzymes that catalyze carboxylic esters to produce an alcohol and a carboxylate. These enzymes participate in phase I metabolism of xenobiotics and in the hydrolysis of long chain fatty acid esters and thioesters (*Ross, Streit & Herring, 2010*; *Lian, Nelson & Lehner, 2017*). CbE activity was measured using ρ-nitrophenyl-α-arabinofuranoside substrate, as indicated by *Hosokawa & Satoh (2001)* with some modifications (25 μL of the supernatant and 200 μL of substrate were mixed and the reaction was recorded for 5 min at 405 nm).

SOD was evaluated using Sigma-Aldrich assay kit (19160), the reaction of which produces a water-soluble formazan dye upon reduction of $O_2^{\bullet-}$. The reduction rate of $O_2^{\bullet-}$ is linearly related to the xanthine oxidase activity of the sample and is inhibited by SOD. CAT activity was measured using *Góth (1991)* method with the modifications made by *Hadwan & Abed (2016)*. In this method, the undecomposed $H_2O_2$ is measured with ammonium molybdate after 3 min to produce a yellowish colour that has a maximum absorbance at 374 nm. Glutathione S-transferase activity was determined from the reaction between GSH and 1-chloro-2.4-dinitrobenzene at 340 nm (*Habig & Jakoby 1981*). Prot were analyzed in the supernatant according to *Bradford (1976)* and was used to normalize enzyme activities. Total GSH was measured with a Sigma-Aldrich Glutathione Assay Kit (CS0260) and was reported in nmol mL$^{-1}$. This kit uses an enzymatic recycling method with glutathione reductase (*Baker, Cerniglia & Zaman, 1990*). The sulfhydryl group of GSH reacts with Ellman's reagent and produces a yellow coloured compound that is read at 405 nm. LPO was evaluated using PeroxiDetect Kit (PD1, Sigma-Aldrich, St. Louis, MO, USA) following manufacturer's instructions and was also reported in nmol mL$^{-1}$. The procedure is based on peroxides oxide iron ($Fe^{3+}$) that forms a colouring component with xylenol orange at acidic pH measured at 560 nm.

## Statistical analysis

With the aim to evaluate physiological condition of females, embryos and paralarvae in an integrative way, several multivariate sets of descriptors were analyzed. Females were first classified following histological criteria developed by (*Avila-Poveda et al., 2016*; *Olivares et al., 2017*): 1) immature (Imm); 2) initial vitellogenesis (stage III of ovocytes) or physiological maturation (Phys Mat); 3) eggs in reproductive coelom or early functional maturation (Ea Func Mat); and 4) eggs at the end of maturation process or late functional maturation (La Func Mat). Female octopus were then analyzed using descriptors of: 1) reproductive condition, that is BW, EBW, OW, RSW, and DG weight; 2) metabolite concentrations in ovaries: Gly, Glu, TG, Chol, and soluble Prot; 3) metabolite concentrations in the DG (Gly, Glu, TG, Chol, Prot), and acid/alkaline protease activities.

*Octopus mimus* embryos and paralarvae were classified considering three well recognized phases during embryo development and two distinct paralarval stages: 1) initial embryonic stages and beginning of organogenesis (Ini Organ); 2) the end of organogenesis with

stages XIV and XV (Organ); 3) stages after organogenesis characterised by body growth (Post); 4) paralarvae with 1 day (1st); and 5) paralarvae with 3 days since hatching (3rd). The descriptors used in the multivariate analysis were: 1) metabolite concentration for the whole egg or paralarvae: Gly, Glu, TG, Chol, Prot; 2) acid and alkaline proteases activity and 3) oxidative stress indicators: redox potential ($E_h$), SOD activity, CAT activity, GST activity, LPO and total GSH; 4) the detoxification indicators, that is activity of AChE and CbE.

The multivariate approaches consisted of ordination by Principal Coordinate Analysis (PCoA) applied on Euclidean distance matrices of samples in each data set (*Legendre & Legendre, 1998*). Data were square root (female data) or log-transformed (embryo and paralarvae data) and normalized by centring and dividing between the standard deviation of each variable prior to analysis (*Legendre & Legendre, 1998*). A permutational multiple analysis of variance (MANOVA) was applied on the distance matrices to detect differences amongst female octopus in different stages of gonadic maturation (fixed factor: Imm, Phy Mat, Ea Func Mat, La Func Mat), and amongst embryos and paralarvae in different stage of development (Ini Organ, Organ, Post, 1st and 3rd day paralarvae). Permutational multiple paired t-tests were used to compare the centroids of the different stages in each data set; 9,999 unrestricted permutations of raw data were used to generate the empirical *F* and *t*-distributions (*Anderson, 2001*; *McArdle & Anderson, 2001*). In an attempt to show data in a more conventional form, the relationship between the descriptors and female's maturation and embryo's developmental stage were also shown as figures with bars without statistical analysis, because such analysis was done using multivariate tests.

# RESULTS

## Female *O. mimus*

Classification of the preserved ovaries of 23 female octopus using histological criteria resulted in eight immature females (Imm) and 15 individuals in different stages of gonadic maturation: four in Phys Mat; six in Ea Func Mat; and five in La Func Mat.

### Reproductive condition

As predicted, the values of all descriptors increased with gonadic maturation, but did so in different proportion. BW and EBW in Ea and La Func Mat stages resulted 3.2 and 2.7 higher than in immature females (Fig. 1A), whereas RSW and OW increased 135 and 108-fold, respectively (Fig. 1A). Increments in the corresponding indices (RSWI and GI) were also observed throughout gonadic maturation (Fig. 1B). Weight of the DG showed its greatest increase (2.6-fold) from Ea to La Func Mat (Fig. 1A). By contrast, the DGI (%) reached a maximum at Phys Mat, that is just before eggs enter the reproductive coelom, but decreased towards the end of La Func Mat (Fig. 1B). Ordination by PCoA of the reproductive descriptors of female *O. mimus* showed 94.9% of total variation explained in the first principal coordinate (PCoA1; Fig. 2A). Both EBW and BW, as well as the OW and RSW were strongly correlated with PCoA1, suggesting they largely contributed to the separation of samples in the horizontal axis. Therefore, samples

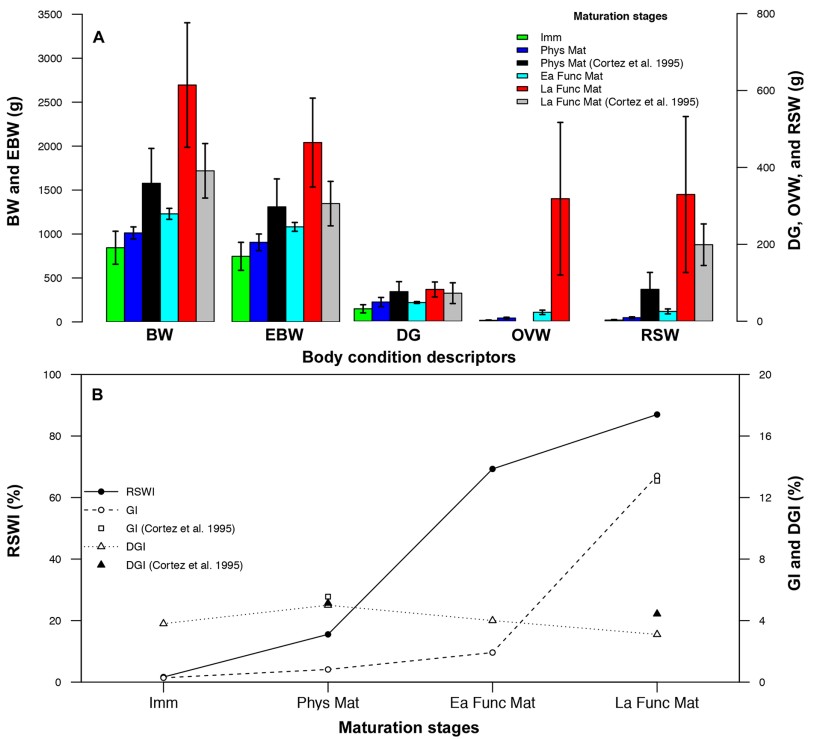

**Figure 1 Body condition descriptors and indices of *Octopus mimus* females along the maturation stages.** (A) Weight (g) of body (BW; left axis), eviscerated body (EBW; left axis), digestive gland (DG; right axis), ovaries (OVW; right axis), and reproductive system (RSW; right axis) in immature (Imm; green bars), physiologically mature (Phys Mat; blue bars), early functionally mature (Ea Func Mat; cyan bars) and late functionally mature (La Func Mat; red bars) females. Body condition descriptors from *Cortez, Castro & Guerra (1995)* are also shown (Phys Mat: Black; La Func Mat: Grey); Values as mean ± SD. (B) Indices (%) of reproductive system weight (RSWI; left axis; black circle and continued line), gonadosomatic (GI; right axis; white circle and dashed line), and digestive gland (DGI; right axis; white triangle and dotted line) along the maturation stages (same as above). GI (white square) and DGI (black triangle) from *Cortez, Castro & Guerra (1995)* are also shown.

with highest values were located at the right-hand side of the ordination map, whereas those with low values were located to the left (Fig. 2A). The second principal coordinate explained only 4.2% of total variation, and was mainly influenced by the weight of the DG. Statistical results obtained with the MANOVA supported these observations since it showed overall significant differences in the descriptors between stages of gonadic maturations (Table 1). In addition, paired comparisons amongst centroids revealed that immature females and those in both Ea and La Func Mat differed significantly in their reproductive condition (Table 2). In summary, immature female *O. mimus* had significantly lower values of all reproductive condition descriptors than those in Ea and La Func Mat stages. Females in Phy Mat stage, however, had statistically similar values (Table 2).

### Metabolite concentrations in the ovaries

Ovary samples of 23 female octopus were used to evaluate metabolites in reproductive tissues. Metabolite concentration in the ovaries of female *O. mimus* changed as gonadic

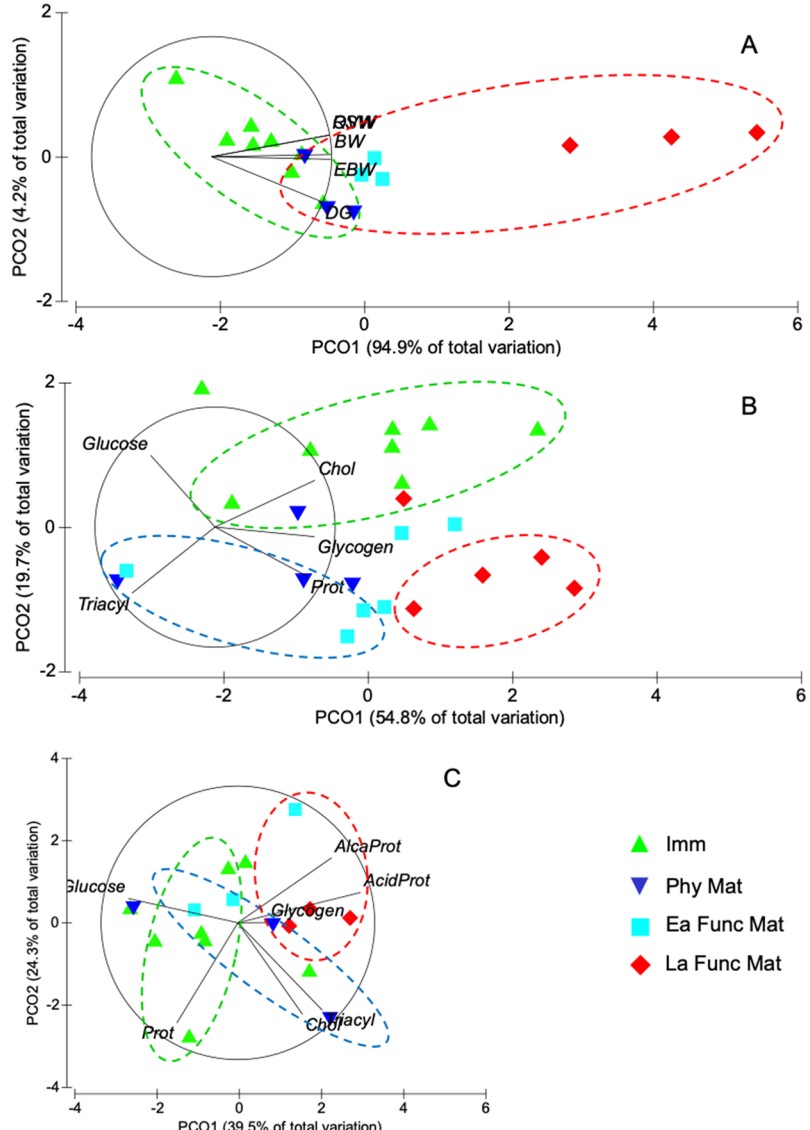

**Figure 2 Principal coordinate analyses (PCoA) of *Octopus mimus* females of different reproductive stages.** PCoA ran with: (A) body condition descriptors: Weight of body (BW), eviscerated body (EBW), digestive gland (DG), ovaries (OVW), and reproductive system (RSW); metabolite concentrations (Glycogen, Glucose, Triacyl = Triacylglycerides, Chol = Cholesterol, Prot = proteins) (B) in ovaries; and (C) in digestive gland together with digestive enzyme activities (Alkaprot: alkaline proteases and Acid-prot: acid proteases). Immature (Imm; green triangle), physiologically mature (Phys Mat; blue triangle), early functionally mature (Ea Func Mat; cyan square) and late functionally mature (La Func Mat; red diamond) females are categorized.

maturation stages advanced, but were not all following the same trends (Fig. 3). Gly decreased from immature ($n = 8$) to Phy Mat females ($n = 4$), and then reached the highest values in La Func Mat octopus ($n = 5$) (Fig. 3A). Glu monotonically decreased, and Prot increased with gained maturity of the females (Figs. 3B and 3C). TG increased from immature females to those in intermediate stages (Phy and Ea Func Mat), but decreased during La Func Mat (Fig. 3D), whereas Chol had the highest values amongst

**Table 1 Results of one-way permutational multiple MANOVAs applied on six multivariate sets of data obtained from *Octopus mimus*: 1) female reproductive condition; 2) metabolite concentration in female ovaries; 3) metabolite concentration and enzyme activity in the digestive gland of female; 4) metabolite concentration; 5) enzyme activity; and 6) antioxidant defence mechanisms in embryos and paralarvae. The degrees of freedom (df), multivariate sum of squares (SS), mean square (MS), pseudo-*F* and *p*-values, and the number of unique permutations is given for each test.**

| Source of variation | df | SS | MS | pseudo-*F* | *p* | Unique permutations |
|---|---|---|---|---|---|---|
| 1. Female reproductive condition | | | | | | |
| Stage | 3 | 70.60 | 23.53 | 32.54 | <0.001 | 9949 |
| Residuals | 13 | 9.40 | 0.72 | | | |
| 2. Metabolites in female ovaries | | | | | | |
| Stage | 3 | 39.66 | 13.22 | 3.57 | <0.01 | 9945 |
| Residuals | 19 | 70.34 | 3.70 | | | |
| 3. Metabolites and digestive enzyme activity in female digestive gland | | | | | | |
| Stage | 3 | 35.52 | 11.84 | 2.01 | <0.05 | 9931 |
| Residuals | 13 | 76.48 | 5.88 | | | |
| 4. Metabolites in embryos and paralarvae | | | | | | |
| Stage | 4 | 202.95 | 50.74 | 19.81 | <0.001 | 9927 |
| Residuals | 75 | 192.05 | 2.56 | | | |
| 5. Digestive enzyme activities in embryos and paralarvae | | | | | | |
| Stage | 4 | 96.88 | 24.22 | 58.66 | <0.001 | 9944 |
| Residuals | 56 | 23.12 | 0.41 | | | |
| 6. Oxidative stress indicators in embryos and paralarvae | | | | | | |
| Stage | 4 | 84.81 | 21.20 | 9.64 | <0.001 | 9928 |
| Residuals | 16 | 35.19 | 2.199 | | | |
| 7. Detoxification indicators in embryos and paralarvae | | | | | | |
| Stage | 4 | 25.98 | 6.49 | 7.42 | <0.001 | 9948 |
| Residuals | 16 | 14.01 | 0.88 | | | |

immature females (Fig. 3E). Ordination of metabolite concentration in the ovaries of female octopus showed that the PCoA1 and PCoA2 explained 74% of total variation in the data (54.8% and 19.2%, respectively; Fig. 2B). Concentration of Gly, Chol, and Prot were positively correlated to PCoA1, hence contributed mostly to the position of samples on the horizontal axis (Fig. 2B). By contrast, Glu and TG were inversely correlated to PCoA2, contributing to situate samples high in Glu in the upper half of the ordination map, and those high in TG at the bottom (Fig. 2B). Results of the statistical procedures applied to the data (Tables 1 and 2) showed significant differences between all stages of gonadic maturation except for Phy Mat and Ea Func Mat females, that were statistically similar (Table 2). In summary, ovary samples from immature females were high in Glu and Chol, but low in TG and Prot, when compared to samples from La Func Mat females; Phy and Ea Func Mat females presented intermediate concentrations of these metabolites in the ovary (Fig. 2B).

**Table 2 Results of permutational paired t-tests that compared centroids representing data in six multivariate sets obtained from female *O. mimus* in different stages of gonadic maturation, and from embryos and paralarvae in different stage of development.**

**1. Female reproductive condition**

|  | Imm | Phy Mat | Ea Func Mat |
|---|---|---|---|
| Phy Mat | 0.053 | – | – |
| Ea Func Mat | <0.05 | 0.102 | – |
| La Func Mat | <0.01 | 0.100 | 0.100 |

**2. Metabolites in female ovaries**

|  | Imm | Phy Mat | Ea Func Mat |
|---|---|---|---|
| Phy Mat | <0.05 | – | – |
| Ea Func Mat | <0.05 | 0.351 | – |
| La Func Mat | <0.01 | <0.05 | <0.05 |

**3. Metabolites and digestive enzyme activities in female digestive gland**

|  | Imm | Phy Mat | Ea Func Mat |
|---|---|---|---|
| Phy Mat | 0.225 | – | – |
| Ea Func Mat | 0.271 | 0.498 | – |
| La Func Mat | <0.01 | 0.204 | 0.104 |

**4. Metabolites in embryos and paralarvae**

|  | Ini Organ | Organ | Post | 1st |
|---|---|---|---|---|
| Organ | <0.01 | – | – | – |
| Post | <0.001 | 0.161 | – | – |
| 1st | <0.001 | <0.001 | <0.001 | – |
| 3rd | <0.001 | <0.01 | <0.01 | 0.567 |

**5. Digestive enzyme activities in embryos and paralarvae**

|  | Ini Organ | Organ | Post | 1st |
|---|---|---|---|---|
| Organ | <0.001 | – | – | – |
| Post | <0.001 | < 0.05 | – | – |
| 1st | <0.001 | <0.001 | <0.01 | – |
| 3rd | <0.001 | <0.01 | <0.01 | 0.617 |

**6. Oxidative stress indicators in embryos and paralarvae**

|  | Ini Organ | Organ | Post | 1st |
|---|---|---|---|---|
| Organ | 0.0915 | – | – | – |
| Post | <0.01 | 0.206 | – | – |
| 1st | <0.01 | 0.096 | 0.103 | – |
| 3rd | <0.01 | 0.101 | 0.099 | 0.099 |

**7. Detoxification indicators in embryos and paralarvae**

|  | Ini Organ | Organ | Post | 1st |
|---|---|---|---|---|
| Organ | 0.605 | – | – | – |
| Post | 0.208 | 0.499 | – | – |
| 1st | <0.01 | 0.103 | 0.200 | – |
| 3rd | <0.01 | 0.099 | 0.097 | 0.302 |

**Note:**
Stages of gonadic maturation (Imm: immature, PhyMat: physiological maturity, EarFuncMat: early functional maturity, and LatFuncMat: late functional maturity), and from embryos and paralarvae in different stage of development (Pre, Organ and Post: stages before, during, and after organogenesis, respectively, and 1st and 2nd paralarvae). Values are permutational *p*-values for each test.

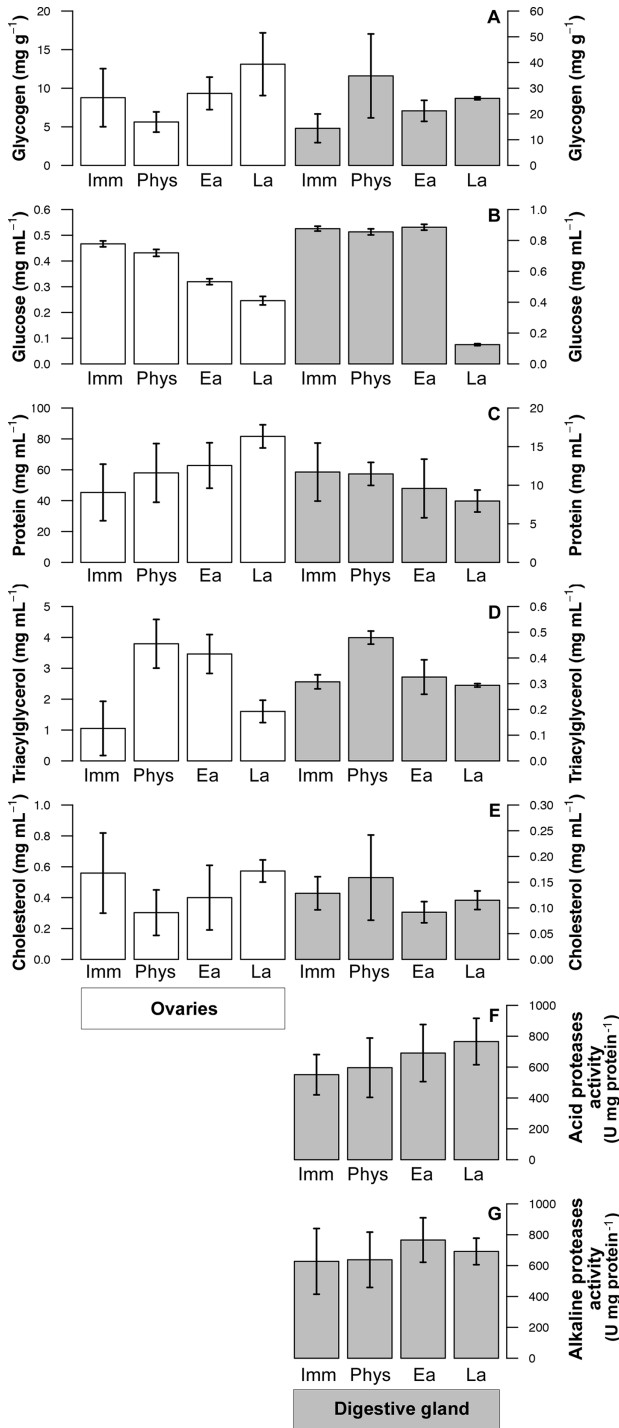

**Figure 3 Metabolites and digestive enzyme activities determined in *Octopus mimus* females of different reproductive stages.** Metabolites were measured in the ovaries (white bars; left axis) and in the digestive gland (grey bars; right axis) of immature (Imm), physiologically mature (Phys), early functionally mature (Ea) and late functionally mature (La) females. The metabolites were: (A) Glycogen (mg g$^{-1}$), (B) Glucose (mg mL$^{-1}$), (C) Protein (mg mL$^{-1}$), (D) Triacylglycerides (mg mL$^{-1}$), and (E) cholesterol (mg mL$^{-1}$). The digestive enzyme activities (F) acid proteases (U mg protein$^{-1}$) and (G) alkaline proteases (U mg protein$^{-1}$) were measured in the digestive gland only. Values as mean ± SD.

### Metabolite concentrations and enzyme activities in the digestive gland

Samples of DG of 23 female octopus were used to evaluate metabolites in this tissue. Levels of Gly were relatively low in the DG of immature female octopus ($n = 8$), but increased 2.4 times in those classified as Phy Mat ($n = 4$) (Fig. 3A). Gly decreased slightly and stabilized in Ea and La Func Mat females. Glu levels in the DG were high and constant in the first stages of gonadic maturation, but decreased markedly in La Func Mat females ($n = 5$) (Fig. 3B). Prot, TG, and Chol levels remained constant throughout maturation stages, but the latter were particularly low in all samples analyzed compared to ovaries (Figs. 3C–3E). The activity of acid proteases increased slightly as females advanced in gonadic maturation (Fig. 3F), whereas alkaline proteases increased from immature females to Phy Mat but were constant thereafter (Fig. 3G). The multivariate analysis of metabolite concentrations and enzyme activities in the DG of female *O. mimus* resulted in the first and second principal coordinates explaining 63.8% of the total variation in the data (39.5% and 24.3%, respectively; Fig. 2C). The percentage of total variation explained by the procedure increased to 79% when the third coordinate was considered. Glu was inversely correlated to Gly, TG, and Chol, whilst Prot were inversely correlated to both acid and alkaline protease activities in all three principal coordinates (Fig. 2C). Significant differences between stages of gonadic maturation were marginally detected by the MANOVA (Table 1). In addition, paired comparisons between centroids only revealed significant differences amongst extreme ends of the maturity gradient (Table 2), indicating that metabolite concentration and activity of proteases in the female DG changed gradually as octopus became mature. It should be noted that values of these descriptors had an overall high dispersion in physiologically mature females, adding to the lack of statistical differences amongst stages (Fig. 2C). In summary, immature females had high concentrations of Glu and, to a lesser extent soluble Prot. La Func Mat females, by contrast had high levels of both acid and alkaline protease activities, but presented low concentrations of Prot and Glu (Fig. 2C).

## Embryos and paralarvae of *O. mimus*

Ten samples of embryos of each stage, and of each paralarvae (1 and 3 days after hatch) of octopus were used to evaluate metabolites, digestive enzyme activities, as well as oxidative stress, and detoxification indicators. When necessary, in the earliest stages, pool of three to five eggs or paralarvae were used.

### Metabolite concentration

Gly levels during the first embryonic stages were relatively similar, but decreased in paralarvae (Fig. 4A). Octopus embryos showed relatively low and constant concentrations of Glu and TG, but these increased in paralarvae (Figs. 4B and 4D). Soluble Prot decreased from the initial to the final embryonic stages, and presented its lowest concentration in paralarvae of the 3rd day (Fig. 4C). Chol, by contrast, stayed in low concentrations during the initial stages of embryonic development, but increased towards the end, showing its highest concentration in paralarvae (Fig. 4E). The first and second PCoA

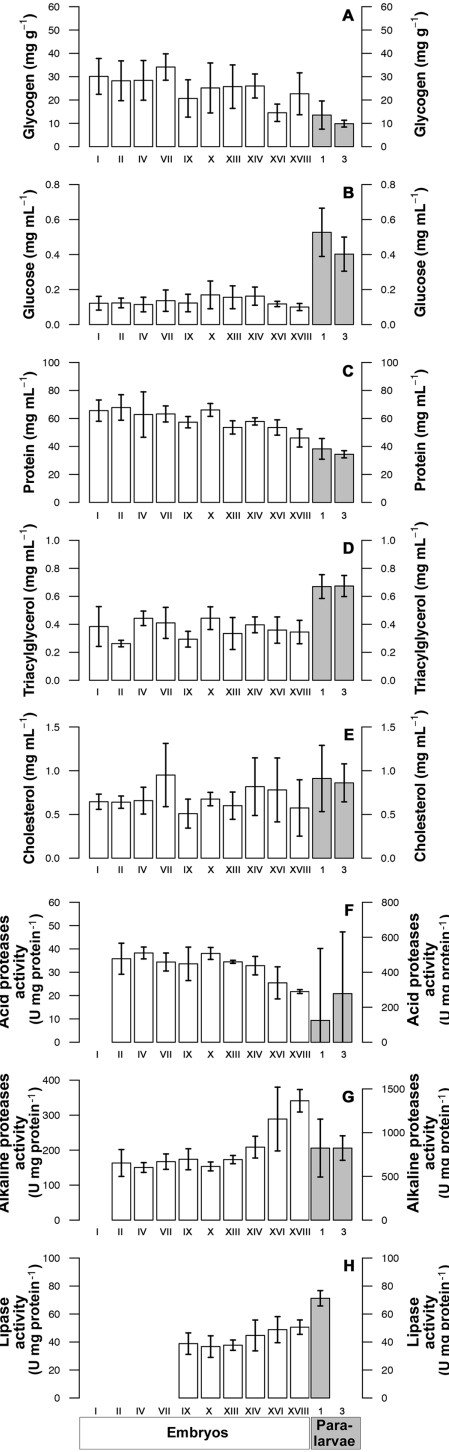

**Figure 4 Metabolites and digestive enzymes determined in *Octopus mimus* embryos and paralarvae maintained at 16 °C in the laboratory.** Metabolites and digestive enzyme activities were measured in ten embryonic stages (white bars; left axis) and in paralarvae of one and three days old (grey bars; right axis). The metabolites were: (A) Glycogen (mg g$^{-1}$), (B) Glucose (mg mL$^{-1}$), (C) Protein (mg mL$^{-1}$), (D) Triacylglycerides (mg mL$^{-1}$), and (E) cholesterol (mg mL$^{-1}$). The digestive enzymes (F) acid proteases (U mg protein$^{-1}$), (G) alkaline proteases (U mg protein$^{-1}$) and (H) lipase were measured in the whole egg or paralarva. Values as mean ± SD.

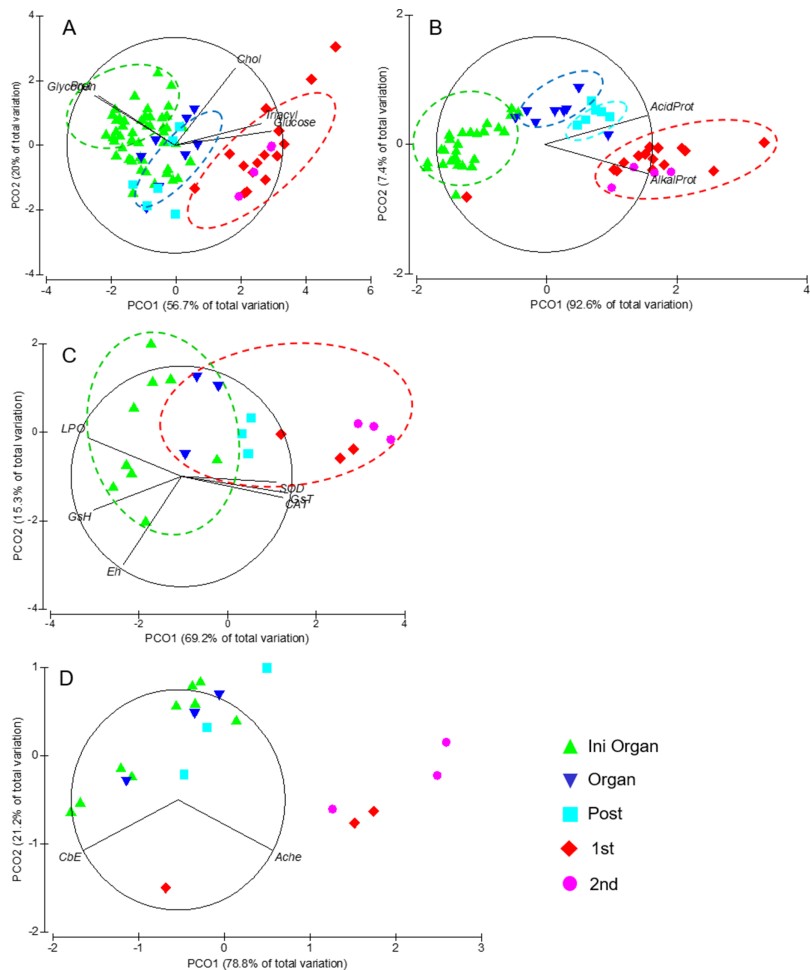

**Figure 5 Principal coordinate analyses (PCoA) of *Octopus mimus* embryos and paralarvae maintained in laboratory conditions (16 °C) at different developmental stages.** PCoA ran with: (A) metabolite concentration (Glycogen, Glucose, Triacyl = Triacylglycerides, Chol = Cholesterol, Prot = proteins); (B) digestive enzymes (Alkaprot: alkaline proteases and Acidprot: acid proteases); (C) oxidative stress indicators: superoxide dismutase (SOD) activity, catalase (CAT) activity, glutathione S-transferase (GST) activity, total glutathione concentration (GSH), redox potential ($E_h$), and lipid peroxidation levels (LPO); and (D) detoxification indicators: acetylcholinesterase (AChE) and carboxylesterase (CbE) activities. Start of organogenesis (Ini Organ; stages I to XII; green triangle), completed organogenesis (Organ; blue triangle), post-organogenesis (Post; cyan square) stages and paralarvae age groups (para-larvae 1: red diamond and para-larvae 3: pink circles) are categorized.

explained 56.7% and 20% of total data variation in metabolite concentrations of octopus embryos and paralarvae (Fig. 5A). Glu and TG were inversely correlated with Gly and Prot and contributed largely to sample ordination on the horizontal axis. Samples high in Gly and Prot were thus located to the left of the ordination map, whereas those high in TG were located to the right. Chol concentration contributed to order samples on the vertical axis (Fig. 5A), so those with high Chol were located at the upper right of the ordination map. The MANOVA detected overall significant differences amongst stages of embryo and paralarvae (Table 1), and significant differences in paired comparisons allowed to distinguish three groups: embryos in stages prior to organogenesis (Ini Organ);

those in stages characterised by organogenesis and immediately after (Organ and Post); and the first and third day's paralarvae. In summary, embryos at the initial phases of organogenesis had high Gly and Prot concentration, but low Glu and TG concentration; embryos in late and post-organogenesis had high Chol concentrations and intermediate values in all other metabolites; 1st and 3rd day paralarvae had high Glu and TG, and low Gly and Prot concentrations (Fig. 5A).

### Digestive enzyme activities

Activity of both acid and alkaline proteases was markedly lower in octopus embryos compared to paralarvae (note the axis different scales for embryos [left] and paralarvae [right]; Figs. 4F and 4G). However, alkaline proteases showed a slight increase in the last embryonic stages. Lipase activity, a yolk consumption proxy, was low in embryos at the initial stages of development, increased towards the end, presenting its highest activity in paralarvae of one day old, and was undetected in paralarvae of 3 days old (Fig. 4H). Ordination of protease activities in embryos and paralarvae of *O. mimus* showed that the PCo1 explained 92.6% of total variation (Fig. 5B), thereby separating samples from left to right in the ordination map (Fig. 5B). Enzyme activities, in general, increased as embryos advanced from initial organogenesis towards paralarval stages, with the highest activity measured in the latter. The MANOVA showed significant change in enzyme activities throughout stages of development (Table 1), with differences detected between all pairs of centroids except those representing the 1st and 3rd paralarvae (Table 2). These results showed four distinct groups of samples regarding digestive enzyme activities: embryos at the beginning of organogenesis, followed by those at the end and post-organogenesis, and paralarvae of days 1 and 3 (Fig. 5B).

### Oxidative stress and detoxification indicators

Concentrations of SOD, CAT, and GST were low amongst embryos in the initial phases of organogenesis, increased throughout development, and attained the highest values in paralarvae (note the logarithmic scale for GST; Figs. 6C–6E). A reduction of oxidative damage was observed throughout development, with LPO and GSH levels starting to decrease at stage XV and attaining the lowest values in paralarvae (Figs. 6F and 6H). In contrast, $E_h$ remained similar throughout embryonic development and the first paralarvae (Fig. 6G). An integrative analysis of the oxidative stress indicators in *O. mimus* embryos and paralarvae showed that the first and second PCoA explained 69.2% and 15.3% of total data variation (Fig. 5C). LPO was high amongst embryos in stages before organogenesis, and was inversely correlated to CAT, GST and SOD, which had the highest values among 1st and 3rd day paralarvae (Fig. 5C). Whilst the MANOVA showed overall significant differences throughout development (Table 1), paired tests amongst centroids revealed significant differences only between extreme stages (Table 2).

Activity of AChE and CbE showed much variation throughout embryonic stages of development (Figs. 6A and 6B). The former, however, showed an important tendency to increase in paralarvae (note the logarithmic scale; Fig. 6A), whereas the later remained relatively constant. Multivariate analysis of the activity of these enzymes showed 78.8%

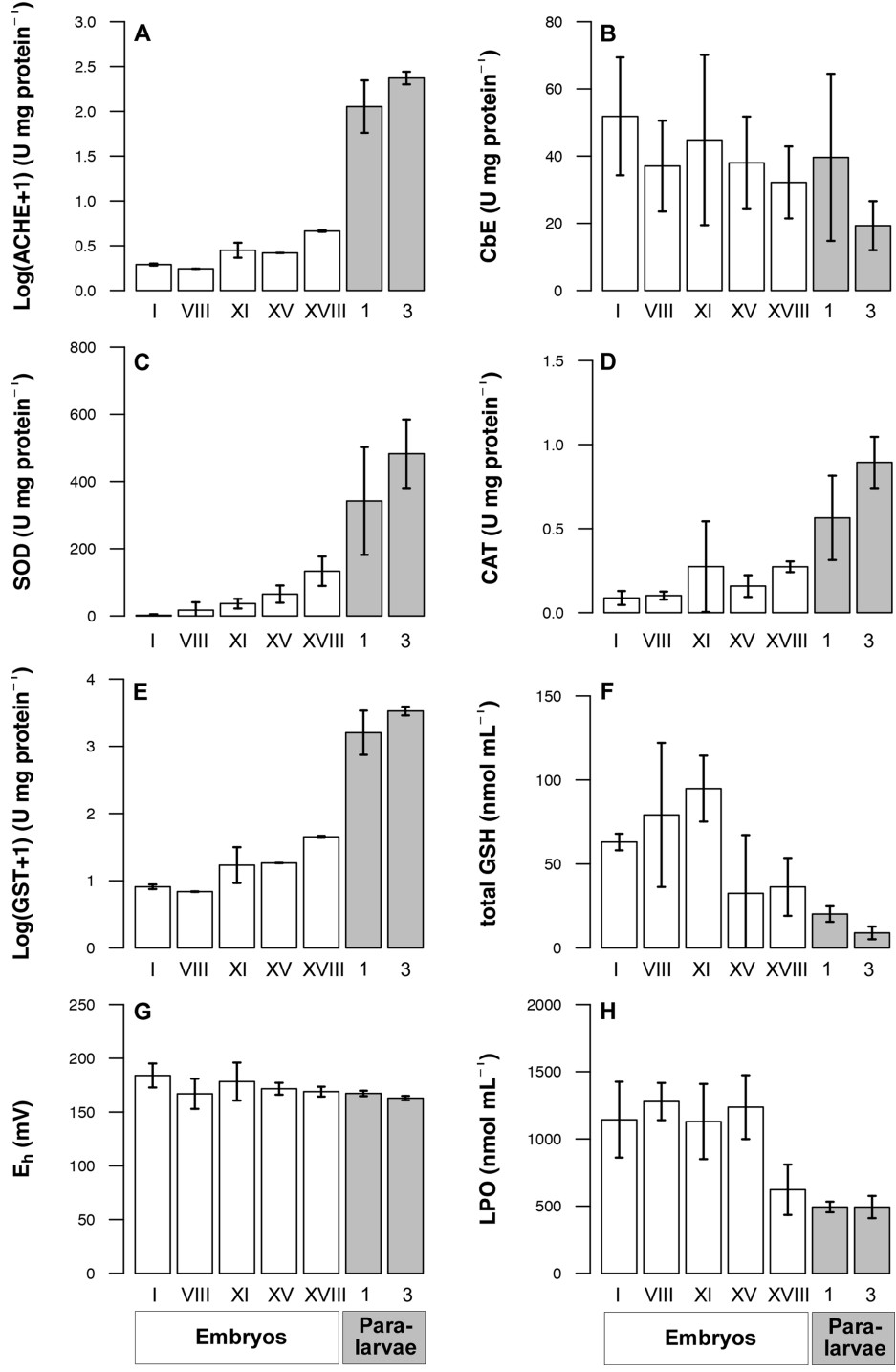

**Figure 6 Detoxification and oxidative stress indicators determined in *Octopus mimus* embryos and paralarvae maintained in laboratory conditions (16 °C).** Indicators were measured in five embryonic stages (white bars) and in paralarvae of one and three days old (grey bars). The indicators were: activities of (A) Acetylcholinesterase (AChE; U mg protein$^{-1}$; logarithmic transformation), (B) carboxylesterase (CbE; U mg protein$^{-1}$), (C) superoxide dismutase (SOD; U mg protein$^{-1}$), (D) catalase (CAT; U mg protein$^{-1}$), (E) glutathione S-transferase (GST; U mg protein$^{-1}$; logarithmic transformation); (F) total glutathione concentration (GSH; nmol mL$^{-1}$), (G) redox potential ($E_h$; in mV), and (H) lipid peroxidation levels (LPO; nmol mL$^{-1}$). Values as mean ± SD.

of variation in the first principle coordinate, with embryo samples with low, and paralarvae with high in AChE activity (Fig. 5D). Activity of CbE, however, only contributed moderately to the ordination of samples. Statistical tests applied to this data showed overall significant differences in AChE and CbE activity throughout stages of development (Table 1), but paired tests amongst centroids revealed significant differences only between extreme stages (Table 2).

## DISCUSSION

The present study provides a strong base line to understand the energy reserves intake and loss during gonadal maturation of *O. mimus* females and the effect of those in embryo and paralarvae characteristics (energetic metabolites, digestive enzymes, oxidative stress/detoxification indicators). Looking at the whole individual organization level, all parameters measured in this study relatively to the weight of digestive and reproductive organs, and their indices, followed the same behaviour as was previously observed when reproductive condition of 991 *O. mimus* females were analyzed (*Cortez, Castro & Guerra, 1995*). Increments on RSW and its RSWI were observed along the reproductive maturity stages, indicating that during ovarian development there were mobilization of energy and nutrients to reproductive organs. Previous studies made in *O. vulgaris*, *O. defilippi* (*Rosa, Costa & Nunes, 2004*) and in the squids *Illex coindetii* and *Todaropsis eblanae* (*Rosa et al., 2005*) suggested that those species take the energy for egg production directly from food rather than from stored products in a specific tissue. That conclusion was based in the observation that, while mature females experience gonad increments on Prot, lipid and Gly contents, the DG and muscle where without apparent changes between both maturation stages, indicating that there were no evidences that storage reserves were transferred from digestive to ovarian tissue during maturation. Although the present study corroborates that conclusion, now we showed that, during maturation process, there were nutrients mobilization from both DG and ovaries that where not observed early. Free Glu in the DG of *O. mimus* females resulted in an important source of metabolic energy, being highly concentrated along the maturation process until Ea Func Mat, when oogonia are growing in the ovaries. In *O. maya*, an elevated level of progesterone (hormone involved in oocytes vitellogenesis) was observed precisely during Ea Func Mat (*Avila-Poveda et al., 2015*), suggesting a high requirement of metabolizable energy to perform this phase. In this sense, previous results obtained in *O. mimus* and *O. maya* showed that DG energy metabolites (Gly, Glu, Prot) were used as a source of metabolic energy during digestive processes (*Gallardo et al., 2017*), and during the growth of juveniles and pre-adults of *O. maya* (*Aguila et al., 2007*; *Rosas et al., 2011*) putting in evidence the importance of those nutrients in the physiology of cephalopods. According to *Martínez et al. (2014)*, this energy is the result of gluconeogenic pathway that are supported by Prot metabolism, that resulted from the carnivorous eating habits of cephalopods. For that reason we hypothesize that, as in DG (*Martínez et al., 2014*), the Gly and Glu encountered in the ovaries followed also the glycogenic pathways (*Hochachka & Fields, 1982*), as previously described in muscle of different cephalopod species (*Gallardo et al., 2017*; *Hochachka & Fields, 1982*; *Morales et al., 2017*).

The dynamics of Gly and Glu observed in *O. mimus* ovaries suggest that there are biochemical regulatory mechanisms involved in the storage and mobilization of nutrients. In the present study high Glu levels were observed in ovaries of immature females and until Ea Func Mat suggesting that Glu could be used as a source of energy during the complex processes involved in oocytes synthesis. However, the reduction of Glu levels recorded in the La Func Mat stage, when the vitellogenesis is at its maximum level, challenges the fact that Glu is the only source of energy. Two hypotheses are proposed to explain this: 1) the latest vitellogenesis taking place in the ovaries may require less energy than previous maturation stages, using facilitating transport mechanisms for vitellogenesis and for storage of yolk into the eggs; or 2) Glu may need to be reduced to avoid inhibitory effects as seen in insects (*Kunkel, 1987*). As Glu in excess could prevents vitellogenin uptake, octopus may maintain minimal Glu levels to be used as a source of metabolic energy, allowing in this form the adequate vitellogenin uptake. If some control mechanisms exist, then Gly synthesis and storage may also be involved. During Ea and La Func Mat, an increment of Gly was registered in ovarium compared to immatures females, suggesting that those molecules were directly stored into the eggs to be used as a source of energy during embryo development and/or to maintain the physiological integrity of females during parental care that occurs after the spawn (*Roumbedakis et al., 2017*).

As was previously observed in *O. vulgaris* (*Rosa, Costa & Nunes, 2004*), no apparent mobilization of lipids from the DG to the ovaries occurred in *O. mimus*, supporting the hypothesis that cephalopods channel nutrients to the ovaries from food intake rather than from tissue reserves, as was also observed in laboratory studies made in *O. maya* and *O. vulgaris* (*Caamal-Monsreal et al., 2015*; *Rosa et al., 2005*; *Tercero-Iglesias et al., 2015*). In the present work, high levels of Chol were registered in immature females, indicating that even before maturation, the females used the ovary as a lipid reserve site. In other words, although it is true that the nutrients come directly from the food and that these are not stored in muscle or DG (*Rosa et al., 2005*), the results obtained now show that the ovary itself is a reserve site for the nutrients that will be used during maturation, at least at the beginning of the process. High Chol concentration may reflect crustaceans ingested during the stage of somatic growth (*Cardoso, Villegas & Estrella, 2004*; *Carrasco & Guisado, 2010*). Once the maturation processes started (Phy Mat), an increment of TGs was observed followed by a reduction of Chol indicating that during oocytes growth, Chol was requested, probably to be used as structural component into the oocytes biological membranes (*Zubay, 1983*). This process is characterised by the formation of oocytes, their growth, and their transformation into secondary oocytes surrounded by the follicle cells without yolk (*Avila-Poveda et al., 2016*). Once the vitellogenesis started (Ea Func Mat), high TG levels were registered, indicating that fatty acids are also stored in the ovaries, probably to be used in the yolk synthesis. The latter is evidenced by low TG levels at the end of the maturation process (La Func Mat) in the ovaries of the females, suggesting their transformation and storage into the eggs as yolk. As Chol accumulated with maturity progression from Phys Mat to La Func Mat, Chol may

also be stored in the yolk to be used by the embryos throughout their development (*Estefanell et al., 2017*).

As can be expected, the biochemical and physiological processes in embryos are highly dynamics following two well identified developmental phases: organogenesis and growth (*Boletzky, 1987*; *Naef, 1928*). In many species, the first phase occurs between stage I to XII-XIII, when the first inversion allows the embryo to growth in the proximal side of the egg (*Boletzky, 1987*). During this phase, nervous system is developed and the retina pigmentation is evident around stage X, to reach the complete organogenesis in *O. mimus* embryos at stages XII-XIII (*Castro-Fuentes et al., 2002*). The organogenesis is characterised by gastrula stages when pairs of arm rudiments are evident joint with first visible organs at the disc-shaped animal pole (*Boletzky, 2003*). Previous experiments in juvenile *O. maya* confirmed that Glu, synthesized via the gluconeogenic pathway, is the final energetic product of Prot catabolism, when the amino acids and polypeptides were transformed into Gly via the gluconeogenesis pathway (*Baeza-Rojano et al., 2013*; *Martinez et al., 2011*; *Rosas et al., 2011*). The present results indicate that soluble Prot and other amino acids could be used, similar to juveniles, as a source of Gly allowing stable and permanent supply of Glu along embryo development, even during the growth phase. In this sense we can hypothesize that Glu is not the most important source of energy for embryos and that regulation of gluconeogenic pathway works as a mechanism for Glu supply. If that regulation exists, it should be coupled with the phases of embryo development being without control until stage XI and control from stage XII onwards. Also, results obtained in the present study putted in evidence that energetic demands of the embryo in the first phase of development were relatively low, without significant mobilization of energetic substrates and its associated enzymes. Also, was registered that eggs were spawned with relatively high levels of LPO and GSH indicating that the maternal ROS production during ovarian maturation was placed in the egg to be eliminated during embryo development. In embryos, the antioxidant defence mechanisms go into action after the stage XIV, when the circulatory systems were activated and an increment on use of yolk as a source of energy was registered (*Caamal-Monsreal et al., 2016*; *Sánchez-García et al., 2017*).

Together with the reduction of soluble Prot and Gly, an increment in lipases activity was recorded in embryos of stage XIV onward, indicating the beginning of yolk consumption. Studies made in *O. maya* and *O. mimus* embryos also found significant mobilization of yolk from stage XIV-XV, indicating stronger mobilization of nutrients from yolk synchronized with embryo growth (*Caamal-Monsreal et al., 2016*; *Sánchez-García et al., 2017*). In *O. mimus*, heart beats increase from stage XV onward, when growth of embryos become evident (*Warnke, 1999*). Exponential growth rate of embryos was obtained from stage XII onwards in *Enteroctopus megalocyathus* (*Uriarte et al., 2016*), suggesting that this phase of growth after organogenesis is a common characteristic between octopus species. The metabolic demand of cephalopod embryos rises during development as was observed in *Sepia officinalis* and *Loligo vulgaris* (*Pimentel et al., 2012*), *E. megalocyathus* (*Uriarte et al., 2016*), *O. vulgaris* (*Parra, Villanueva & Yúfera, 2000*) *O. mimus* (*Uriarte et al., 2012*) and *O. maya* (*Caamal-Monsreal et al., 2016*;

*Sánchez-García et al., 2017*). In parallel with this higher metabolic demands, ROS are produced, which consequently leads to oxidative stress if not eliminated by the antioxidant defence system (*Regoli et al., 2011*). A recent study on the role of the enzymatic antioxidant system in the thermal adaptation of *O. vulgaris* and *O. maya* embryos suggests that early developmental stages of cephalopods have temperature-regulated mechanisms to avoid oxidative stress (*Repolho et al., 2014*; *Sánchez-García et al., 2017*). In the present study, it was observed that ROS in the eggs of *O. mimus* embryos were controlled during the growth phase of the embryo development. This was thanks to the activation of the antioxidant defence mechanisms in the stage XIV, indicating the coupling between metabolic demands and the functioning of the antioxidant defence system against oxidative stress. The GST particularly increased in the paralarval stages, which could explain the lower levels of GSH in those stages as this is its substrate. It has also been found that AChE plays an important role during morphological modification (*Fossati et al., 2013*), supporting the hypothesis that AChE participates in the regulation of cell proliferation and apoptosis (*Robitzki et al., 1998*). The important increase of AChE in the paralarval stages could be the result of this transition between embryos and paralarve stages. In *O. maya* embryos, a marked increase in AChE at the beginning of the organogenesis has also been reported (*Sánchez-García et al., 2017*). No trends were observed regarding CbE activity in *O. mimus* embryos, which speaks for little role in yolk catabolism.

Studies made in *O. vulgaris* hatchlings revealed that during the burst of anaerobic swimming of paralarvae the energy is obtained from Glu and from Arginine phosphate system mediated by lactate dehydrogenase and octopine dehydrogenase respectively (*Morales et al., 2017*). Both systems require pyruvate, either by the gluconeogenic route or via the degradation of amino acids by transamination (*Zubay, 1983*). Although we do not have data to demonstrate what kind of mechanisms could be operating in *O. mimus* embryos, it is possible to hypothesize that those mechanism should be related with amino acids and lipid catalysis, because both substrates (mainly arginine and glycerol) are involved in pyruvate production in cephalopods (*Morales et al., 2017*). In this sense we think that undetected activity of lipases until stage IX, and the moderated activity of alkaline and acidic enzymes registered along *O. mimus* embryo development could be involved in physiological regulation of the energy supply in those organisms. Studies made in *O. maya* embryos demonstrated that the yolk consumption starts after the stage XIV, when the organogenesis ends (*Caamal-Monsreal et al., 2015*; *Sánchez-García et al., 2017*), just when the antioxidant defences are activated.

## CONCLUSION

Results obtained in the present study demonstrate that ovarium is a site for the reserve of some nutrients for reproduction. Presumably, TGs are stored at the beginning of the maturation processes together with Chol; we hypothesize that both were energetically supported by Glu, derived from Gly following gluconeogenic pathways. These findings suggest the existence of a control mechanism of Prot-Gly-Glu operating in *O. mimus*' ovarian. We suggest that Glu, while is the energetic support for ovarian maturation at the

beginning, could have a potential role as an inhibitor of vitellogenin uptake at the end of the maturation processes. Results obtained in the present study suggest that ROS produced during the metabolic processes that occur in ovarian maturation were in part transferred to the egg, provoking a ROS maternal charge to the embryo. The elimination of this ROS maternal charge in the embryos started when the activity of the heart is initiated and the higher absorption of the yolk is noted around stages XIV and XV. After stage XV of development, the activation of the energetic metabolism and digestive enzymes, joint with increments in consumption of Gly were also observed. All these processes could be key in preparing suitable antioxidant defence mechanisms for the paralarvae that are obligated to neutralize ROS production when they start their zooplankton life style, experiencing an increase in energetic requirements related to food acquisition.

### Funding

This study was supported with funding from the Programa de Apoyo a Proyectos de Investigación e Innovación Tecnológica of the Universidad Nacional Autónoma de México (PAPIIT-UNAM) IN219116 awarded to Carlos Rosas and partially financed by DGECI through the collaboration network of TEMPOXMAR. The Dirección General de Asuntos del Personal Académico-UNAM provided a Postdoctoral position to Nelly Tremblay, and the Consejo Nacional de Ciencia y Tecnología (CONACYT) provided an infrastructure I010/186/2014 grant to Carlos Rosas. There was no additional external funding received for this study. The funders had no role in study design, data collection and analysis, decision to publish, or preparation of the manuscript.

### Grant Disclosures

The following grant information was disclosed by the authors:
Programa de Apoyo a Proyectos de Investigación e Innovación Tecnológica of the Universidad Nacional Autónoma de México (PAPIIT-UNAM) IN219116.
Partially financed by DGECI through collaboration net of TEMPOXMAR.
The Dirección General de Asuntos del Personal Académico-UNAM.
Consejo Nacional de Ciencia y Tecnología (CONACYT): I010/186/2014.

### Competing Interests

The authors declare that they have no competing interests.

### Author Contributions

- Alberto Olivares conceived and designed the experiments, performed the experiments, contributed reagents/materials/analysis tools, authored or reviewed drafts of the paper, approved the final draft.
- Gabriela Rodríguez-Fuentes performed the experiments, contributed reagents/materials/analysis tools, prepared figures and/or tables, authored or reviewed drafts of the paper, approved the final draft.
- Maite Mascaró analyzed the data, contributed reagents/materials/analysis tools, prepared figures and/or tables, authored or reviewed drafts of the paper, approved the final draft.
- Ariadna Sanchez Arteaga performed the experiments.
- Karen Ortega performed the experiments.
- Claudia Caamal Monsreal performed the experiments, authored or reviewed drafts of the paper.
- Nelly Tremblay performed the experiments, prepared figures and/or tables, authored or reviewed drafts of the paper, approved the final draft.
- Carlos Rosas conceived and designed the experiments, performed the experiments, analyzed the data, contributed reagents/materials/analysis tools, prepared figures and/or tables, authored or reviewed drafts of the paper, approved the final draft.

## Animal Ethics

The following information was supplied relating to ethical approvals (approving body and any reference numbers):

The actual study was approved by the Animal Care committee of the Universidad de Antofagasta, Chile, following the approved Experimental Animal Ethics Committee of the Faculty of Chemistry at Universidad Nacional Autónoma de México (Permit Number: Oficio/FQ/CICUAL/100/15).

## Data Availability

Olivares, Alberto; Rodríguez-Fuentes, Gabriela; Mascaró, Maite; Sánchez, Ariadna; Ortega, Karen; Caamal-Monsreal, Claudia; Tremblay, Nelly; Rosas, Carlos (2018): Energetic metabolites, digestive enzymes, antioxidant defense mechanisms (REDOX) and radical oxygen species (ROS) of *O. mimus* females and embryos. Universidad Nacional Autonoma de Mexico, PANGAEA, DOI 10.1594/PANGAEA.889686

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
