# Peer review of "Maturation trade-offs in octopus females and their progeny: energy, digestion and defence indicators"

_PeerJ, doi:10.7717/peerj.6618_

## Round 0.1 · original submission · Major Revisions

Please address the issues raised by all three reviewers. Please pay close attention to the presentation and interpretation of results and discussion section.

Reviewer 1 ·

Basic reporting

The MS requires much work to present clearly both methods and results for several reasons detailed below and in the pdf. Also, the introduction and discussion need to be improved.

There is objectives defined.
The presentation of the paper looks like a draft. The format of the figures, abbreviation, spelling errors, etc. ... that should be reviewed deeply, there are many errors that makes it very difficult to review the work.

Authors should also revise all the MS, and the English should be improved, please see some examples in the detailed comments in the pdf.

Experimental design

no comment

Validity of the findings

no comment

Additional comments

This study performs analyses the biochemical changes during sexual maturation and its effects on embryonic development of O. mimus, an important octopus specie in the Pacific Ocean. It examines the levels of metabolites, digestive enzymes and antioxidant defences system in females during gonadal maturation, and during embryo development.

Although this is not the first study showing changes in biochemical composition related with sexual maturation, and/or during embryos development in cephalopods, including octopus, this paper is interesting because provides new information on for this octopus species, about how energy and antioxidant defences can change during sexual maturation (in specific tissues) and embryonic development.

But the MS requires much work to present clearly both methods and results for several reasons detailed below and in the pdf. Also, the introduction and discussion need to be improved.

The introduction would benefit from expanding the general ideas given in several paragraph (ej. Lines 69-- “ A mixed maternal diet (what do you mean?) resulted in more hatchlings (what do you mean? ) from O. vulgaris females than from females fed with mono-diet (only crabs) (Márquez et al. 2013. Lines 78-84 “Caamal-Monsreal et al., (2016) and Sánchez-Garcia et al., (2017) showed that from stage XV onwards………these studies are related to Tª, then the paragraph must refer to this abiotic factor.). Also, it is important to state the objectives.

Another concern is the format of the figures, abbreviation, spelling errors, etc. ... that should be reviewed deeply, there are many errors that makes it very difficult to review the work.

The discussion would greatly benefit if the author explains why some enzymatic activities in particular the AChE, CbE and GST, are determined and discussed together with antioxidant enzymes.

Another concern is the format of the figures, use of different abbreviations or names to describe the same thing, spelling errors, etc. ... that should be reviewed deeply, there are many errors that makes it very difficult to review the work.

Authors should also revise all the MS, and the English should be improved, please see some examples in the detailed comments in the pdf.

Title:

The title does not reflect properly the content of the paper, which is manly biochemistry analysis. Rewrite a title that better fits the study.

Annotated reviews are not available for download in order to protect the identity of reviewers who chose to remain anonymous.

Reviewer 2 ·

Basic reporting

I’m not a native speaker but I think that the English language is clear and technically correct. The authors have shown a good knowledge of the previous researches. They have a wide experience in this field with several papers in nutrition and metabolism in cephalopods.
In general terms, introduction is well focused, although some aspect should be improved: Other comments about the introduction: Line 75-76: I think the introduction arguments about females and eggs should be split in two different paragraphs. How did you select the antioxidant mechanisms and ROS to be analyzed? A paragraph about the role of Ache, CbE, ORP, etc should be included. Authors should consider to put the paragraph 93-116 at the beginging of the introduction. The last paragraph of the introduction is not well connected with the previous paragraph. Other references to be considered on this field
Steer, M.A., Moltschaniwskyj, N.A., Nichols, D.S., Miller, M., 2004. The role of temperature
and maternal ration in embryo survival: using the dumpling squid Euprymna
tasmanica as a model. J. Exp. Mar. Biol. Ecol. 307, 73–89.

Farías, A., Navarro, J.C., Cerna, V., Pino, C., Uriarte, I., 2011. Effect of broodstock diet on the
fecundity and biochemical composition of eggs of the Patagonian red octopus
(Enteroctopus megalocyathus Gould 1852). Cienc. Mar. 37 (1), 11–21.
Cagneta, P., Sublimi, A., 1999. Productive performance of the common octopus (Octopus
vulgaris C.) when fed on amonodiet. Seminar of the CIHEAMNetwork on Technology
of Aquaculture in the Mediterraean on “Recent Advances in Mediterranean Aquaculture
Finfish Species Diversification”.
Quintana, D., Márquez, L., Arévalo, J. R., Lorenzo, A., & Almansa, E. (2015). Relationships between spawn quality and biochemical composition of eggs and hatchlings of Octopus vulgaris under different parental diets. Aquaculture, 446, 206-216.
Zuniga, Oscar & Olivares, Alberto & Rosas, C. (2014). Octopus mimus. Cephalopod Culture. 397-413. 10.1007/978-94-017-8648-5_21.
The structure present a standard format and the manuscript could be considered as an "unit of publication" including all the relevant results according to the hypothesis proposed. From my point of view, most of the figures are relevant, although a deep review of the details should be made. There is an important lack of description in the figures as well as heterogeneity of information among different figures. The legend should include a general title or description and then a detailed description of each figure (A, B, etc.). On the contrary, detailed description of some parameters (ex. maturation stages in Fig 1 and 2 or embryonic stages in Fig 8) should be avoided, because has been described in Mat and Met. All legends should include the meaning of all abbreviations. You must use the same denomination of parameters (Example to avoid: reproductive system weight in Mat and Met vs Reproductive complex system in Results vs reproductive tract in fig 1). Fig 1 and 2, Cortes or Cortez? The statistical letter are not always well allocated Ex. Fig 4). The different statistical comparisons should be indicated in the footnote (Ex . a vs a’ vs a’’ in Fig 4). Legend of Fig 3 doesn’t include information about the three graphs (A, B and C). Fig 3 (B). uncorrected values in the glucose axis. Embryonic stage should be named as roman numbers ( I, II, IV…). Fig 4. Axis legend only mention acid protease. In summary authors should revise and homogenise the information among material and methods, figures and results because in the current form is confusing and with a lot of misinformation.
Regarding the Figure 9 and 10, I agree with the aim of the authors to summarize the results. However, in the present form, the real contribution of these figures is questionable, particularly for Fig 9. The pathways proposed by the authors (Ex. Protein to glycogen, TG to Cho, etc.) are not the only option and further studies are needed to confirm the prevalence of these pathways. I agree to propose a theoretical model but the authors should highlight that it is just a hypothetical model. If Fig 9 cannot be improved according to these comment, its should be eliminated. The theoretical model could be including in the text. Regarding Fig 10, I’m not sure if it is worthy to be include. In any case, I would change the “XXX” by “+++”. I think that the development of each parameter should be include in the text instead of the figure legend.

Experimental design

The research is within the aims and scope of the journal. The knowledge gaps are correctly identified and the hypothesis and methodology to fill the gap are well elaborated. Both ethical and technical standards are high. The methodology description is enough to be reproducible although a clearer scheme of the sample analysis is necessary. There are many different analyses in this manuscript and the current exposition is confusing about this aspect. For future studies, I recommend to use MgCl2 or ethanol for humanitarian end-point according to the Fiorito et al., 2015 and Butler-Struben et al., 2018.
Fiorito, G., Affuso, A., Basil, J., Cole, A., De Girolamo, P., D’angelo, L., ... & Mark, F. (2015). Guidelines for the Care and Welfare of Cephalopods in Research–A consensus based on an initiative by CephRes, FELASA and the Boyd Group. Laboratory animals, 49(2_suppl), 1-90.
Butler-Struben, H. M., Brophy, S. M., Johnson, N. A., & Crook, R. J. (2018). In vivo recording of neural and behavioral correlates of anesthesia induction, reversal, and euthanasia in cephalopod molluscs. Frontiers in physiology, 9, 109.
Some minor aspects should be improved in Material and methods section. All the devices should include name, country and city of the company.Line 141. Eviscerated weight is the fifth weight instead of four. 143. Oviductal gland weight is separated in Fig 1. 174. I understand that no food was supplied to the paralarvae but this aspect should be mention. “mL” instead of “ml”. Acylglicerides are not exactly TG (Include MG and DG). It can be assumed that most of them are TG and you can use both terms but always the same. Anyway you cannot use acylglicerides levels as lipid levels because phospholipids has not been measured. In similar ways Bradford is a technique for soluble protein, It is not a good indicator for several structural protein such as collagen. 193-196. How did you decide the Ph of acid and alkaline proteases? Previous studies should be mentioned. Please include details about “ Universal buffer”. There are not trypsin and lipase data at the first egg stages and 2 days-old paralarvae. Pkease, explain the reason. 219-223. Samples were frozen in liquid nitrogen, lyophilized, stored at -20ºC and later, supernatant stored at -80ºC. Are you sure that enzyme activity was maintain at -20ºC despite of the lyophylized process? 274-276. Why don’t you name initial (Ini) organogenesis instead of Pre organogenesis? 278-280: abbreviations should be explained. Lipase and trypsine should also be mentioned

Validity of the findings

No previous data about this subject have been published in Octopus mimus. Since paralarval viability is the most important bottleneck in octopus aquaculture, to improve the hatchling quality through broodstock nutrition and embryonic development is a very useful research line in this field. The data obtained are robust and statistically sound. There is an adequate number of replicates and the sample timing is correct to study the metabolism development in the digestive gland, ovarium and eggs.
However, several aspects of the results should be revised. 309-312. Please revised this text according with Fig 2. 316. The beginning of the maturation process is the “ Phys Mat or “Ea FuncMat”? What is the difference between this sentences and the next sentence (2.4 times higher). The data and the name (HI%) are different between them. 320-321: A reduction in 38% of glycogen. When? 365: What about the CHO in the eggs? 369-379: please review carefully this paragraph according to the Fig 5 there are several mistakes. 385. C to E instead of B to D. 385. Why don’t you consider CbE as antioxidant enzymes instead of oxidative damage? 401. Which differences? 405: 19.7 according to the table. 420: a negative correlation. 426-427. Tese results are different from the results described in Fig 5 and line 374-376. It should be eliminated. 437-438. Glu, Cho and TG are similar in all egg stages (see figure). This text 436-441 should be eliminated. 447. Similarly “lower acid protease than embryos” is not correct.
Regarding the discussion and the conclusion, the authors should make a deeply review of this section on the basis of capacity of the techniques used in the present study. As I mentioned before, I agree with these techniques to obtain a general overview of the nutrient metabolism, but to go beyond, additional techniques such as radiolabelled studies, gene expressions or intermediate metabolic enzymatic profile should be used. In addition some relevant nutrients such as phospholipids, free aminoacids or non soluble proteins have not been determined. My point of view is that several assessment about nutrient sysnthesis or metabolism are highly speculative because are based just in the nutrients contents in different tissues along the development. Therefore the conclusions should be adapted to the methodology limitations. Authors have include new data/references in Conclusion section (ex. 623-626). I think that these references belong to the discussion section. Conclusion should be a general overview without detailed comments. Maybe a brief summary of the hypothetical models explained in the figures 9 and 10.
Minor changes: 475-479. Both sentences appear to be in contradiction between them. Please review them. 559-561. I cannot understand this sentence

Reviewer 3 ·

Basic reporting

no commnet

Experimental design

The experimental design is adequate and the methods described are in general with sufficient detail, except some questions explained below. However, the study of additional ages of post-hatching paralarvae would also be included helping to reinforce the results and discussion described.

Validity of the findings

no comment

Additional comments

The study reports the biochemical composition changes that take place during gonadal maturation of Octopus mimus females and its consequences in embryo and hatching paralarvae. This physiologic study is very interesting and helpful for the improvement of the rearing of paralarvae of O. mimus, a species with a high potencial for aquaculture in the Pacific Ocean area. The results obtained are adequate and very interesting. However, as main comment, the conclusions concerning the biochemical and physiological dynamics that occurs in female maturation and through embryo development and post-hatching are quite speculative. Authors should perform additional analysis at biochemical and even at molecular level to study additional enzymes and molecules involved in the referred signaling pathways in order to ascertain the activation of that biochemical pathways.
Some specific suggested changes and comments are reported below:
-Line 101, 105: Please, change "68 d old" by "68 days old". Check it all around the manuscript.
-Line 173: Please, specify the protocol of sampling method of embryos. Was the whole embryo frozen?, or do the authors proceed to dechorionation of eggs before freeze the samples for posterior biochemical analysis?. It is also important to know if the results obtained are from embryos with or without yolk sac.
- Line 174: Please, explain this sentence. "Paralarvae of 1 or 2 days post-hatching were sampled and analyzed at biochemical level". What is the reason that authors decided to sampling at that sampling points so close?. No big differences are described between 1 and 2 days post hatching in paralarvae, and in fact, the statistical results reported detected significant differences among stages, but no differences among 1day and 2 days post hatching paralarvae were observed. However, a sampling point at more advanced age would be of great interest.
- Line 315 and Fig. 3A. The standard deviation of the data of glycogen obtained in female DG (and in some other data presented in the manuscript) are very wide and therefore the results and conclusions obtained should be checked.
-Line 317: Please, change "Fig. 1A" by "Fig. 3A".
-Line 320: Please, rephrase or explain in better way: "A reduction in 38% in digestive... values was registered".
-Line 359: Data of cholesterol in embryo and paralarvae corresponding to Fig. 4B were not included in the results section. Please, add it.
-Figure 3: Please, add "3A", "3B" and "3C" and indicate it with the corresponding letters in the graph.
-Figure 4: Same thing than Fig.3
-Figure 5: Please, add "5A" an "5B" to the figure legend.
-Figure 8. This figure is not clear in the current format. Letters are overlapping and are not read properly

---

## Round 0.2 · Minor Revisions

Please incorporate minor changes suggested by Reviewer 2 on your manuscript.

Reviewer 2 ·

Basic reporting

No comments

Experimental design

No comments

Validity of the findings

No comments

Additional comments

The qulity of the manuscript has been clearly improved, however, there are still some minor changes

Abstract:
Line 26: A list of metabolites (as in line 31) should be included
33-49: This paragraph include some statements that cannot be not fully proved with the techniques carried out in the present study (I will give further details onwards). I consider that authors should start with a sentence like this: “ Based on the data obtained, we hypothesised that….”. In general terms, I agree with the interpretation of the data, but it should be clear that several aspects of this paragraph are speculation or are not fully proved (ex. Cho storing, gluconeogenic pathway, energetic cost of the organogenesis, maternal origin of the ROS in the egg or ROS production during planktonic life). This comment is also valid for the Conclusion of the manuscript.
Introduction:
72-73: Marquez et al., 2013 used three mono-diets instead of mixed diets. Crab and squid produced better results than fish. Please review the paper and Quintana et al., 2015 Aquaculture 446: 206-216.
Results:
Several statements should be revised because there is no enough statistical support to maintain them. These statements are mainly related with the changes in metabolites or enzyme activity along the development. Most of the time, there are too much variability to obtain clear conclusions. See details below:
Section 3.1.2. Line 368: Given the SD, there is no a clear reduction of Gly between “Imm” and “Phys” (Fig 3A). This sentence should be eliminated. Line 373: Cho levels of “La Func” are very similar to “Imm”. Line 384: Authors cannot say that “Imm” are higher in Cho and lower in TG than “La Func” . See Fig 3.
Section 3.1.3: Lines 392-394: Protein is the only metabolite that seems to reduce in the “La Func” stage and despite of that it is not clear if this difference are significative. In similar ways (line 396). Alkaline proteases don’t seem to increase in a significant way. I think that the sentence between 408-410 ( variability in “Phys” stage) should be located among the first sentences of this section. Line 411 and 412: I don’ agree about these comments regarding “Prot” and alkaline protease.
Section 3.2: This section should include a initial paragraph similar to section 3.1 with details about the number of the samples for the analysis, as well as the specific stages of the development used. For example, Which specific stages were selected for “Ini Organ” or “Post” groups? How many samples from each stages?
Section 3.2.1: Line 420-422, 427-429 and 423: I cannot see a clear tendency in the Cho levels. This data should be rewrite taking into account the high variability of the Cho. Line 427: “…high in TG and Glu..”
Section 3.2.3: Line 463: “ (Fig. 6G).” Line 464: Review the % data. Are different from the figure. Line 465: “among”
Discussion:
Line 487: A reference for this study is needed
Line 545-554: The high variability of the Cho should be mention in this paragraph. I agree with the interpretation, but in the present study there is no enough data for this statement. This is a speculative hypothesis.
Line 576-582: I don’t understand how the authors obtain this conclusion. Please include more details and argument to justify this paragraph.
Line 589 and 593: Usually, I understood that embryos used the yolk from the beginning of the development. Why do you use the word “Yolk” just from organogenesis onwards?
Conclusion: As I mentioned before some statements are not fully proved. You cannot use some words as “This study demonstrate” (line 643) , “indicate” (line 651) or “these process were key” (line 656) among others. Please, focus this aspects as speculative arguments.
Table 2: Section 3. “Metabolites and digestive enzyme activity in female…”

---

## Round 0.3 · accepted · Accept

The revised manuscript is acceptable in the present form.

#